# Deep learning exoplanets detection by combining real and synthetic data

**Sara Cuéllar[1]☯, Paulo Granados[1]☯, Ernesto Fabregas[2], Michel Curé[3], Héctor Vargas[1‡], Sebastián Dormido-Canto[2‡], Gonzalo Farias[1]***

**1** Escuela de Ingeniería Eléctrica, Pontificia Universidad Católica de Valparaíso, Valparaíso, Chile,
**2** Departamento de Informática y Automática, Universidad Nacional de Educación a Distancia, Madrid, Spain,
**3** Instituto de Física y Astronomía, Facultad de Ciencias, Valparaíso, Chile

☯ These authors contributed equally to this work.
‡ HV and SDC also contributed equally to this work.
* gonzalo.farias@pucv.cl

**Data Availability Statement:** All databases are public.

**Funding:** This research was supported in part by the Chilean National Agency for Research and Development (ANID) under Projects FONDECYT

## Abstract

Scientists and astronomers have attached great importance to the task of discovering new exoplanets, even more so if they are in the habitable zone. To date, more than 4300 exoplanets have been confirmed by NASA, using various discovery techniques, including planetary transits, in addition to the use of various databases provided by space and ground-based telescopes. This article proposes the development of a deep learning system for detecting planetary transits in Kepler Telescope light curves. The approach is based on related work from the literature and enhanced to validation with real light curves. A CNN classification model is trained from a mixture of real and synthetic data. The model is then validated only with unknown real data. The best ratio of synthetic data is determined by the performance of an optimisation technique and a sensitivity analysis. The precision, accuracy and true positive rate of the best model obtained are determined and compared with other similar works. The results demonstrate that the use of synthetic data on the training stage can improve the transit detection performance on real light curves.

## Introduction

All the planets in our solar system orbit the sun. Planets orbiting other stars are called exoplanets under NASA's Exoplanet Exploration Program [1].

Exoplanets are very difficult to see directly with telescopes. They are hidden by the brightness of the star they orbit. The search for planets outside the solar system has been investigated for many years. The existence of a possible exoplanet orbiting the white dwarf Van Maanen 2 has been suspected since 1917 [2], but its existence could not be confirmed due to the limited technology of the time.

It was not until 1995 that Michel Mayor and Didier Queloz first confirmed an exoplanet called Dimidium or 51 Pegasi, with a 4-day orbit around the nearby star Helvetios [3]. They described it as a large ball of gas similar to Jupiter. For this finding they received the Nobel Prize in Physics 2019 [4].

1191188 and 1190486, and PhD Scholarship 21221393. The National University of Distance Education under Project 2021V/-TAJOV/00 and Ministry of Science and Innovation of Spain under Project PID2019-108377RB-C32.

**Competing interests:** The authors have declared that no competing interests exist.

Nowadays, scientists and astronomers have attached great importance to the task of discovering new exoplanets, even more so if they are in the habitable zone. Most of the exoplanets discovered so far are found in a relatively small region of our galaxy, the Milky Way. To date, NASA has confirmed 4301 exoplanets, using a variety of discovery techniques [5], including planetary transits, radial velocities, gravitational microlensing and direct imaging from databases provided by space and ground-based telescopes, e.g. NASA's Kepler space telescope [6] and the NASA's Transiting Exoplanet Survey Satellite (TESS) [7].

The Kepler space telescope has collected data on a large number of stars (in the order of 200,000) during the 4 years it was operating (2009-2013). Collecting around of 678 Gigabytes of data [8]. As many other examples in the literature (see for instance [9–12]), the manual analysis of large databases, such as these light curves, is a very time-consuming work. In this context, the use of artificial intelligence methods have emerged as tools for the analysis of this information. The main research questions addressed by this work are how can artificial intelligence algorithms contribute to the exoplanet detection field, and if it is possible to add technical knowledge through synthetic data to improve the performance of the detector.

In the literature, different approaches that use artificial intelligence techniques to detect exoplanets can be found. For example, in [13], the authors describe a method for detecting exoplanet transits by applying the k-nearest neighbors (kNN) method to determine whether a given signal is sufficiently similar to known transit signals. In [14], they present for the first time the use of the Random Forest Classifiers (RFCs) algorithm for exoplanets classification. They achieve an overall error rate of 5.85% and an error rate in the classification of exoplanet candidates of 2.81%. The work described in [15], shows a combination of RFCs and Convolutional Neural Networks (CNNs) to distinguish between the different types of signals. The authors say that the combination of both methods offers the best approach to identify exoplanets correctly in the test data approximately 90% of the time. While in [16], the authors present another CNN based approach that is capable of detecting Earth-like exoplanets in noisy time series data with a greater accuracy than a least-squares method. The most important disadvantage of this case is that they use synthetic data to train the model instead of real traffic data. This does not provide evidence for its performance against real data.

In [17], the method for classifying candidates using a Self-Organizing Maps (SOM) technique is developed on Kepler and K2 confirmed and candidate planets with a success of 87%. More recently, in [18] an Ensemble-CNN model for exoplanets detection is presented with an accuracy of 99.62%.

Other approaches such as [19], shows a 98% cross-validated precision score using RFCs to classify objects of interest in Kepler's cumulative information object table. But, in this case, the authors use only data from the training stage for cross-validation of their models. This does not allow to properly analyse the performance of the model with new data.

Despite the good results obtained by these previously mentioned works, most of them show that in order to build and validate the models, in some cases light curves of unconfirmed planet candidates are used or even some of them are false positives. The main contributions of this work are the following:

- The development of a system for detection of planetary transits in Kepler Telescope light curves which includes the generation of synthetic data from estimated parametric models of the planet candidate. This approach allows finding planetary transits over a wider range of periods.

- As far as we know our approach is the first exoplanet detection model trained by deep learning from a mixture of real and synthetic data. A sensitivity analysis and an optimisation technique is performed to determine the best ratio of synthetic data. The model consists on

building an image from the folding of light curves. This image is used to determine planetary transits by means of a CNN.

- Unlike other related works, the validation of the model is only performed with real data and different from those used in the training stage. This shows that the performance of the model is better than if only real data are used for training.

This paper is structured as follows. Second section presents some exoplanet detection approaches that can be found in the literature and describes briefly the approach which is the start point of this work. Third section details the proposed method. Fourth section shows the experimental results and a comparison with previous results. Finally, Fifth section summarises the main conclusions and future work.

## Exoplanets detection approaches

As mentioned above, the discovery of new exoplanets has taken a high degree of importance during the last few years. Since the amount of data provided by telescopes is enormous, the only way to analyze it is using Machine Learning techniques. A significant amount of research can be found in the literature that has focused on the use of Machine Learning techniques for exoplanet detection. This section presents a review of the most significant and relevant works for our approach.

Table 1 presents a summary of the articles covered in this brief review. The first column contains the reference to the article in the bibliography. The second column shows the names of the telescope and catalogs from which the data were obtained. The third column shows the details about the feature extraction used. The fourth column shows the machine learning method used for detection. Finally the fifth shows the results obtained by each approach.

In [20], published in 2015, the authors present the *Autovetter*, a machine learning based classifier. It is used to produce a catalog of Planet Candidates from the Q1-Q17 DR24 Threshold Crossing Events (TCEs) that are identified in the Kepler Science Operations Center pipeline. The *Autovetter* classify 20367 TCEs into three classes: 1.- Planet Candidate (PC), which contains 3600 signals that are consistent with transiting planets; 2.- Astrophysical False Positive (AFP), which contains 9596 signals of astrophysical origin that could mimic planetary transits; and 3.- Non-Transiting Phenomenon (NTP), which contains 2541 signals that are evidently of instrumental origin, or are noise artifacts. A set of 114 atributes calculated from

**Table 1. Machine learning approaches for exoplanet detection.**

| Ref | Catalog | Feature Extraction | ML Method | Performance |
|---|---|---|---|---|
| [20] | REAL Kepler Q1-Q17 DR24 | 114 Attributes calculated | RF (3 classes) | Accuracy: 0,973 |
| [21] | REAL Kepler Q1-Q17 DR24 | 1D folding curve: global & local view | LLR Fully connected NN CNN | Accuracy: 0.917, 0.94, 0.958 AUC: 0.963, 0.977 0.988 |
| [22] | REAL Kepler Q1-Q17 DR24 | 1D folding curve: global & local view Centroid curves Stellar parameters | DCNN | Accuracy: 0.975. Precision: 0.955 |
| [23] | REAL TESS 1-5 sector | 1D folding curve: global & local view Secondary eclipse view | CNN for Triage | Accuracy: 0.974. AUC: 0.992 Precision: 0.97 |
| [19] | REAL Kepler Cumulative | Features from interactive table | SVM, KNN, RF | Training metrics Accuracy: 0.9896. Precision: 0.9955 Recall: 0.9721 F1: 0.9837 |
| [24] | SIMULATED with tansit REAL without transit | 50000 lightcurves: 25000 with transit 25000 without transit | MLP, CNN | Accuracy: 0.99. Recall: 0.99 |
| [25] | SIMULATED REAL Kepler Q1-Q17 DR24 TESS 1-5 sector | TSFresh 789 features | Gradient Boosted trees | Simulated AUC: 0.92 Recall: 0.92 Precision:0.94 Kepler AUC: 0.948. Recall: 0.96 Precision:0.82 TESS AUC: 0.80. Recall: 0.82 Precision:0.81 |

Kepler pipeline are ultimately used to build a random forest classifier that maps the attributes of any TCE to a predicted class label of either PC, AFP, or NTP. The results evaluated on 4630 TCEs show the following accuracy/error rate for each class: PC (0.971/2.9%), AFP (0.976/ 2.4%) and NTP (0.968/3.2%). As can be seen, these results are very accurate, in fact, the *Auto-vetter* predictions are taken as ground truth for posterior studies.

In [21], published in 2018, the authors present a method for classifying potential signals from planets using deep learning, specifically convolutional neural networks (CNNs). Feature extraction is generated by folding each flattended light curve in the TCE period (with the event centered) and clustering to produce a 1D vector. The training and test sets (PC, AFP and NTP) were selected from the *Autovetter Planet Candidate Catalog* for Q1-Q17 DR24. The result is a CNN model named *Astronet* that is able to distinguish with good accuracy the subtle differences between genuine transiting exoplanets and false positives such as eclipsing binaries, instrumental artifacts, and stellar variability. They also compared models based on linear logistic regression (LLR) and a fully connected neural network. The results show a performance of classified real planets with 95% recall, 90% of accuracy and 96% of precision.

In [22], also published in 2018, the authors also present an approach based on CNN named *Exonet*. They use a dataset from the same catalog as the previous one (Kepler Q1-Q17 DR24). For the classification process, they use phase-folded light curves and associated centroid curves (measured by the Kepler pipeline from the same TPF), for both global and local views. They also add stellar normalized parameters like: effective temperature, surface gravity, metallicity, radius, mass, and density to the training set. The results overperformed the *Astronet* with an accuracy of 97.5% and 95.5% of precision.

In [23], published in 2019, the first deep neural network trained and tested on real TESS data is presented. The model is modified based on *Astronet* and designed to automatically performing triage and vetting on TESS candidates. In triage mode, it can distinguish transit-like signals (planet candidates and eclipsing binaries) from stellar variability and instrumental noise with an average precision of 97.0% and an accuracy of 97.4%. In vetting mode, the model is trained to identify only planet candidates with the help of newly added scientific domain knowledge, and achieves an average precision of 69.3% and an accuracy of 97.8%.

In [19], also published in 2019, the authors present a study of several classification models (SVM, KNN and RF) used to assign a probability of an observation being an exoplanet. A Random Forest Classifier was selected as the optimum machine learning model to classify the data on the Cumulative Kepler Object of Interest (KOI) catalog, which contains information for all Kepler Objects of Interest (KOI) in one place. The Random Forest Classifier, trained using the table attributes as features, obtained a cross-validated accuracy score of 98.96%, precision 99.55% and recall of 97.21% on the training set.

In [24], published in 2019, the authors present an approach based on CNN for detecting exoplanet transits. A 2D phase folding technique is proposed, generating a set of images for training. They test the method with five different types of deep learning models with or without folding. Synthetic lightcurves were generated as the input of these models. The results indicate that a combination of two-dimension convolutional neural network with folding is the best choice for the future transit analysis. All models with folding have accuracy above 98%. The accuracy of models without folding can become about 85%. The precision and recall have a similar trend. This article is based on this approach, the main difference is that it uses real data with transit for both training and testing.

In [25], published in 2020, the author present an approach based on a tree-based classifier using a popular machine learning tool *lightgbm*, to detect exoplanets using the transit method. They use time-series analysis library *TSFresh* to extract 789 features from lightcurves. These features capture information about the characteristics of each lightcurve. This method was

trained and tested on synthetic data and real Kepler and TESS data. The evaluation on synthetic data proved it to be more effective than conventional box least squares fitting (BLS). On Kepler data, the method is able to detect a planet transit with an AUC of 94.8% of accuracy and Recall of 96%. With the TESS data, the method is able to classify lightcurves with an accuracy of 98% and is able to identify planets with a Recall of 82%.

## Proposed method

This section describes the entire process of building the model, from the data collection, the application of the feature extraction process, and the methodology for the training and validation stages of the proposed approach.

### Real data description

The dataset consists of Kepler observations of near 200,000 stars started from 2nd May 2009 to 11th May 2013. The data is divided in 18 quarters from Q0 to Q17. The length of each quarter is about 90 days, but some quarters are shorter. The data includes long and short cadence which took data every 30 and 2 minutes respectively. Long cadence data will be only considered from Q1-Q17 quarters because there are not enough stars observed in short cadence.

The Transit Planet Search (TPS) module carefully observes the light curves and identifies possible signals called Threshold Crossing Events (TCE). The Data Validation module creates reports based on the probability of veracity of the signals; then the *Robovetter* [26] examines the signals and creates a Kepler Objects of Interest (KOI) catalog. Those confirmed to have nothing to do with planetary transits are labeled as false positives (FP). The remain are called planet candidates (PC). NASA provides the list of all confirmed planet transits (CP) as well the planet and star properties.

The Cumulative Kepler Objects of Interest (KOI) table provides the most accurate dispositions and stellar and planetary information for all KOIs in one place. The KOI catalog table contains unique object of interest identifiers, exoplanet archive information, transit properties, among others threshold-crossing events properties. The labels were sourced from the catalog's *koi_disposition* column as the ground truth. The catalog contains 9564 KOIs, out of which 2358 are confirmed exoplanets, 2366 remain candidates and the rest (4840 objects) are false positives. The last group of objects was removed from the dataset. Since since it is searching for signals with planet transits, the candidates and confirmed exoplanets have both been combined into the transit labelled data presented in the next sections.

The non-transit labelled data were obtained from the Kepler Data Release 25-Q1 (DR25) table and consists of 43273 KOIs with no transit.

Two observed fluxes columns are used from each observation data: one is the simple aperture photometry (SAP) which is the flux obtained by direct photometry analysis and can include some other device; the other one is the Pre-search Data Conditioning (PDC) which represents a processed version of SAP where the devices are removed almost completely [27]. PDC light curves will be used. Since the unusual values produced by astrophysics events such as solar flares and micro lenses are not eliminated of the PDC light curve, all the points over 6 times standard deviation will be removed.

Since many of confirmed planets share the host star with other planets, systems with only one confirmed planet or PC was chosen. Following the approach from [24], the main idea is to use a light curve with enough samples to represent 10 periods, considering that the bigger transit period, the higher amount of samples. At Kepler observations case, a curve has about 4320 samples taken each 30 minutes then the maximum allowed period to cover the 10 segments is 9 days. Therefore all transits with a period between 0.85 and 8.5 days were considered. On this

point there are 583 light curves of the Q1 quarter, in order to maintain balance, the same amount of non-transit light curves was selected.

## Synthetic data generation

The method proposes to include a set of transit synthetic light curves in order to improve the performance of the classifier by taking advantage of the technical knowledge acquired from experts of the periodicity and modelling of the planet's transit. Following the approach proposed on [24], those are generated using the quadratic model for the limb darkening laws introduced analytically by Mandel & Agol on [28]. The flux $f$, for a transit over a stellar disk with quadratic limb darkening is:

$$f(k, z) = 1 - \frac{(1 - c)\lambda_e(k, z) + c\lambda_d(k, z) + u_2\eta_d(k, z)}{1 - u_1/3 - u_2/6}, \tag{1}$$

where $k$ is the radius ratio, $z$ is the projected distance, $c = u_1 + 2u_2$, $u_1$ and $u_2$ are the quadratic limb darkening coefficients, and $\lambda_e$, $\lambda_d$ and $\eta_e$ are functions of $k$ and $z$ defined in [28]. This model is implemented on the PyTransit Python library [29] with related parameters like the transit period $\tau$, the ratio of planet radius to stellar radius ($r_p/r_s$), the ratio of orbital semimajor axis to stellar radius ($a/r_s$) and the orbit inclination ($i$). The values of those parameters were set the same as [24], Table 2 gives a summary of them.

## Feature extraction

**Lightcurves pre-processing.** The main purpose of this method is to find transits, so the light curves are pre-processed to first remove the empty intervals [24]. The missing values are replaced with the average of the neighborhood of the empty interval. Noise with less than 10% of magnitude is added on this new values for a better consistency with reality. The light curves were interpolated to 4000 points i.e. 10 periods of 400 points each. Finally the data is normalized to have values between 0 and 1.

**2D phase folding.** Related work on light curve feature extraction includes the phase folding technique introduced by [16] to take advantage of transit periodicity. It consists on folding each light curve on the transit period from the catalog and binning it to generate a 1D vector of a enhanced signal. This method increase the transit detection but the transit period has to be known in advance, otherwise the folding period can differ from it and the transit will be undetectable for the model. The above condition represents a difficulty when it is wanted to search for transits on new released observational data. To solve this problem [24] presents a phase folding method that generates a 2D representation by folding each light curve on a period that can be different from the transit period, improving the transit detection regardless of the transit and folding period. Following this method, the detection model inputs were

**Table 2. Transit parameters.**

| Parameter | Value |
|:---:|:---:|
| $\tau$ | 0.85 to 8.5 [days] |
| $a/r_s$ | 2 to 35 |
| $r_p/r_s$ | 0.005 to 0.4 |
| $i$ | 85 to 90 [deg] |
| $u_1$ | 0.210 to 0.731 |
| $u_2$ | 0.035 to 0.442 |

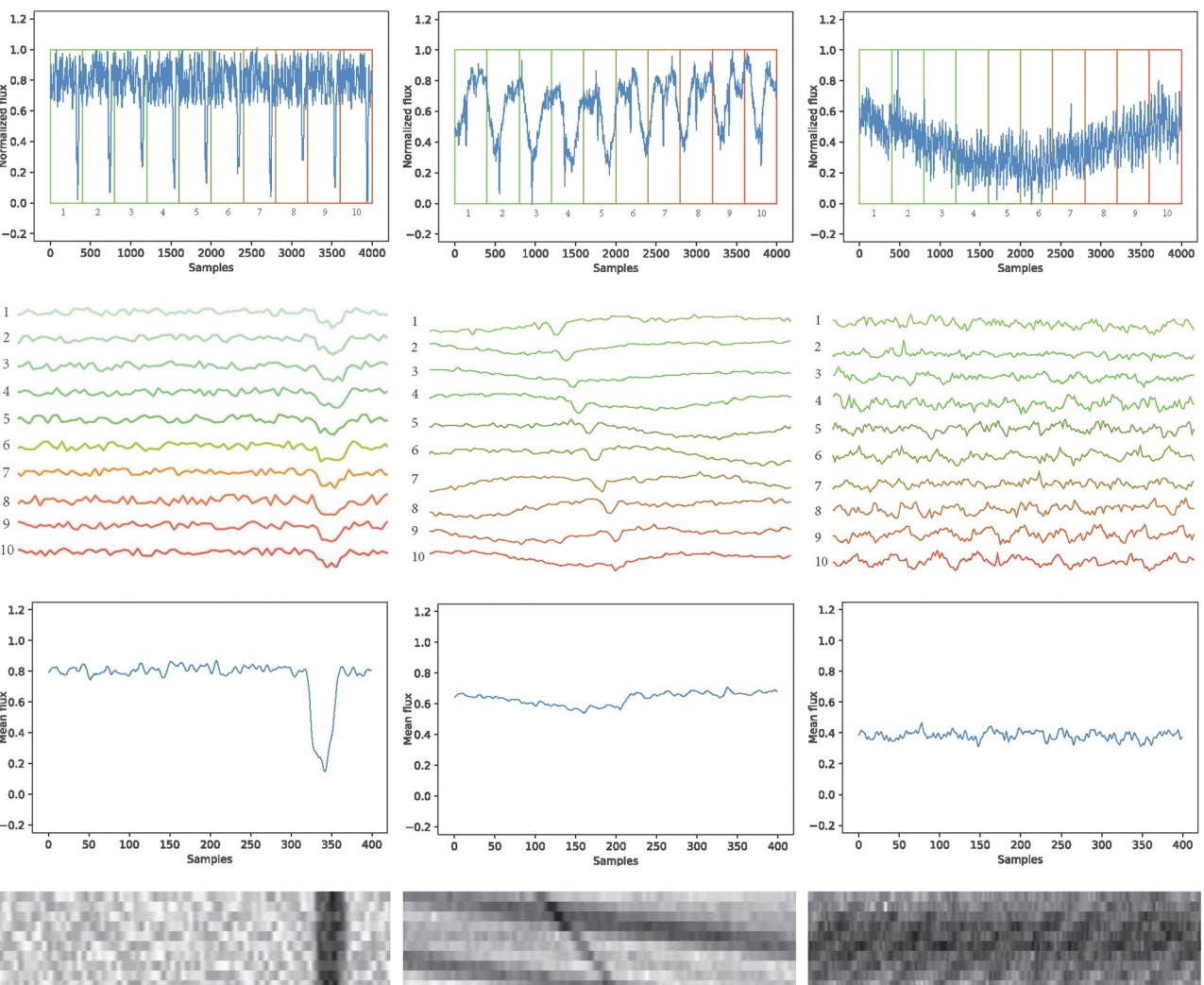

**Fig 1.** 2D Phase folding for three cases: Synthetic lightcurve (left). Real lightcurve with transit from the host star *KIC7051180* (center). Real lightcurve without transit from the host star *KIC757076* (right).

generated by folding the light curve in 10 segments using the transit period from the catalog's *koi_period* column, then the values for each period were incorporated as rows of an image.

Fig 1 shows the 2D phase folding process on three examples of lightcurves: first column (left) shows a synthetic lightcurve with transit folded on the transit period, second column (center) shows a real lightcurve with unsuitable folding for the actual transit period, and third column (right) presents a real lightcurve without transit. First row of the figure presents the pre-processed light curve, a 1D signal of 4000 samples normalized to have values between 0 and 1. Ten folds of 400 samples each are enumerated. Second row of the figure shows a zoom view of each fold, the transit is visible on (left) at the same phase on each fold, since (center) has a folding period different from the transit period, the drops on each fold appears to be shifted to a lightly different phase than in the previous one, on (right) no transit drop is visible on any fold. Third row of the figure presents the mean of the 10 folds on each case, this would be the inputs of the [16] proposed model, i.e. a 1D vector of 400 samples. The transit is visible on (left) closer to the 350 sample, on (center) the mean transit signal becomes unclear and the

model may not give a correct answer on transit detection, confusing it with the one presented in (right). Finally, fourth row shows the 2D representation of the folded lightcurve, a 10x400 pixels image which is the input of the detection model. The dark bands indicating the transit are visible on (left) and (center). Therefore, a model may be able to detect transits successfully and distinguish it from the one in (right).

### Detection model

The classification task consists on sort out every light curve into two categories: Planet candidate and False positive. Different kind of deep learning approaches have been used for this application. In our approach, a convolutional neural network was chosen since they outperform artificial neural networks in classification tasks where the data is spatially aligned such as image or audio [30]. It is because CNN leverages the spatial structure of the output detecting local features which only need to be learned once, therefore the number of trainable parameters, the memory usage, and the number of computations of the desired output will decrease.

Transfer learning takes a large network that has already been trained for a specific problem and then fits it to a new problem. This adjust is performed at the end of the network, modifying the number of output neurons to match the number of classes of the new problem (2 classes). This is a very useful technique since the first stages of the network usually recognize general features that can be applied to almost every classification problem [31]. Clearly it is necessary to perform train in order to adapt the last layer to the new classes, but thanks to transfer learning it is not necessary to train the whole network again. In fact, one can choose which layers to train and which not to train. This is very efficient when considering the computational cost of training a network of this magnitude.

The proposed convolutional neural network architecture is based on *Xception* developed by Francois Chollet. This CNN has learned rich feature representations from the ImageNet dataset [32], outperforming Inception V3 on it (which Inception V3 was designed for) and significantly outperforms Inception V3 on a larger image classification dataset comprising 350 million images and 17,000 classes [33]. Xception is a linear stack of depth-wise separable convolution layers with residual connections that contains 71 deep layers. It can classify 1000 categories of objects and has an image input size of 299x299. It is necessary to modify the input dimension of the network and the number of output neurons. In this binary classification problem a monochromatic image input size of 400x10 and one neuron on the output layer with a sigmoid function of activation were implemented. The output $y$ of the model depends on the neural network decision threshold $T$ (where $0 < T < 1$). This threshold determines the minimum classification probability on which the light curve will be classified as a planet candidate (the probability predicted is greater than $T$) or as an false positive (the probability predicted is smaller than $T$) as shown Eq 2 where $z$ is the weighted sum on the inputs.

$$y = \begin{cases} 1 & \text{if } \dfrac{1}{1 + e^{-z}} \geq T \\ 0 & \text{otherwise} \end{cases} \tag{2}$$

### Training the model

In order to see the effect of increasing synthetic data on training, DCNN models were built with $R = 483$ real curves and a ratio of synthetic curves $S$ with transit defined by the $\lambda$

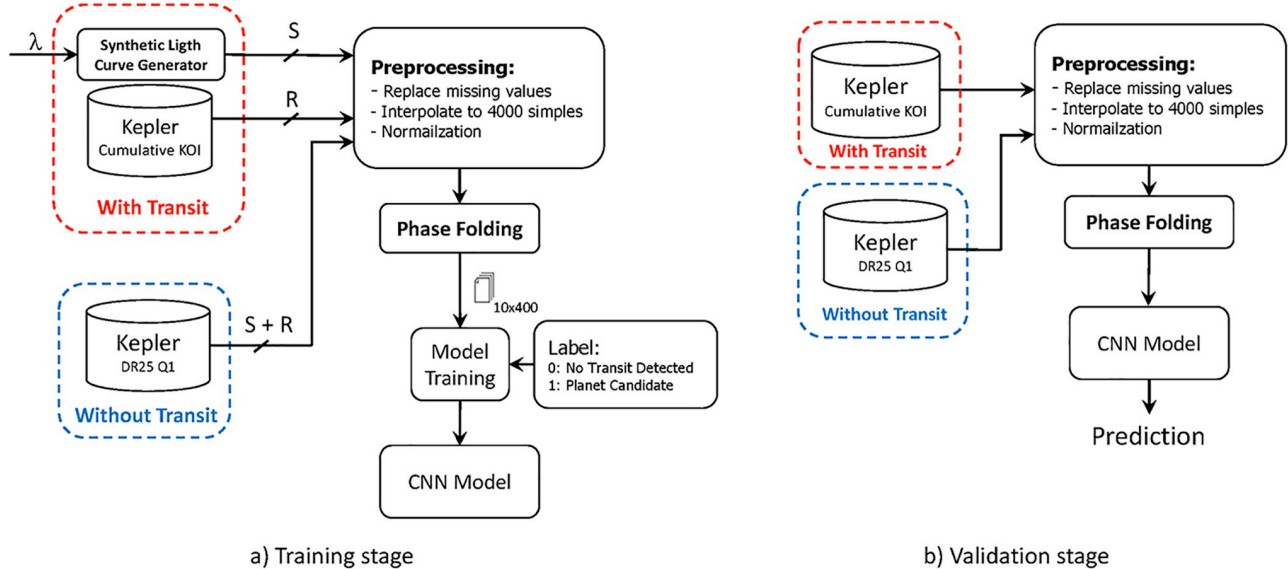

**Fig 2. Training and validation stages of the proposed method.**

parameter (see Eq 3).

$$S = \frac{\lambda R}{100 - \lambda} \quad (3)$$

Additionally, the same number $S + R$ of non transit curves are added to the training set to maintain balance. Fig 2(a) summarizes the workflow from the training stage.

## Evaluation metrics

For evaluation purposes, we select as test set: $R = 100$ real transit light curves and $R = 100$ non-transit light curves. This work uses the following metrics to asses the performance of the CNN model based on the workflow shown in Fig 2(b):

- Accuracy: The portion of correct classifications.

$$accuracy = \frac{TP + TN}{TP + FP + TN + FN} \quad (4)$$

- Precision: The ratio of lightcurves classified as planet candidates that are true planet candidates, also known as reliability.

$$precision = \frac{TP}{TP + FP} \quad (5)$$

- True Positive Rate (TPR): The ratio of true planet candidates that are classified as planet candidates, also known as recall.

$$TPR = \frac{TP}{TP + FN} \tag{6}$$

- False Positive Rate (FPR): The ratio of non transit lightcurves misclassified as planet candidates.

$$FPR = \frac{FP}{FP + TN} \tag{7}$$

- Finally, $F_1$-Score is calculated as shown in Eq 8 and is used to evaluate in a single value the combination of both precision and recall.

$$F_1 = 2 \times \frac{precision \times recall}{precision + recall} \tag{8}$$

## Experimental results

The values of the accuracy, precision and TPR depend on the the neural network threshold $T$ chosen for the model see Eq 2. The main hypothesis of this work is that the performance of the model can be also improved using a ratio $\lambda$ of synthetic lightcurves on the training stage. This section presents the ratio selection of synthetic lightcurves using a coarse and a fine tunning, also a comparison between the proposed method and the ones reported in the literature is presented.

### Best $\lambda$ ratio selection

The detection performance was tested on a test set of $R = 100$ real curves with transit and the same amount of real curves without transit, on two extreme scenarios:

1. Training with only real lightcurves with transit ($S = 0$, $R = 483$) and $S + R$ real lightcurves without transit.

2. Training with only synthetic lightcurves with transit ($S = 483$, $R = 0$) and $S + R$ real lightcurves without transit.

On both scenarios the neural network threshold is fixed on $T = 0.5$ due to the fact that this is a binary classification problem where 0 is a predicted lightcurve without transit and 1 is a predicted planet candidate.

Table 3 shows the performance of the models built for each scenario. It is observed that the model trained with only real curves can detect the 74% of the transit light curves from the test

**Table 3. Detection performance under proposed scenarios.**

|  | With transit | | Without transit | |  |  | Metrics | |
|---|---|---|---|---|---|---|---|---|
| Scenario | S | R | S | R | T | $F_1$ | TPR | Precision |
| 1 | 0 | 483 | 0 | 483 | 0.5 | 0.743 | 0.740 | 0.747 |
| 2 | 483 | 0 | 0 | 483 | 0.5 | 0.206 | 0.120 | 0.750 |

set with a precision of 74.7%; this is a good rate however underperform the reported metrics from the literature. On the other hand when the model is trained with synthetic curves it can detect the transit lightcurves with a similar level of precision than the model trained with real curves, but clearly it does not have all the variability of curves with transit.

Therefore, it can be concluded that training with real curves provides variability to the model on transit lightcurves detection but incorporate synthetic light curves can improve the precision of the prediction. In order to find the best ratio of $\lambda$ and $T$, a coarse tunning with an heuristic optimization method and a sensibility analysis are proposed.

## Heuristic search of optimal parameters

The optimization problem is described by Eq 9, the amount of synthetic lightcurves $S$ and the neural network threshold $T$ are the decision variables.

$$\underset{S,T}{\text{maximize}} \quad F_1(\lambda, T)$$

$$\text{subject to} \quad 0 \leq \lambda \leq 80\% \tag{9}$$

$$0 < T < 1$$

The fitness function is the balanced $F_1$-score, which is the harmonic mean between precision and TPR calculated on Eqs 5 and 6 respectively where:

- True Positive (TP): lightcurve with transit detected as planet candidate.

- False Positive (FP): lightcurve without transit detected as planet candidate.

- True Negative (TN): lightcurve without transit detected as false positive.

- False Negative (FN): lightcurve with transit detected as false positive.

Genetic algorithms have been widely used in the last decades, because they are considered a tool to solve complex optimization problems managing the influence of the uncertainties of typical design engineering scenarios. The main idea behind GA is to evolve a population of chromosomes (possible candidate solutions of the problem), in several iterations (also called generations), using operators such as crossover and mutation and evaluated under a fitness function. In this context, this article uses GA as an optimization tool to find a small range to narrow down the search for the values of $S$ and $T$ in order to obtain the highest possible $F_1$ value. Algorithm 1 shows the pseudo-code of the implemented GA, and Table 4 summarizes the parameters settings used for the GA implementation.

**Table 4. GA implementation details.**

| Parameter | Description | Value |
|---|---|---|
| Encode $S$ | Integer | 11 bits |
| Encode $U$ | Two decimals | 7 bits |
| Chromosome size | Encoded $S$ and $U$ concatenated | 18 bits |
| Population size | Number of chromosomes in one generation | 10 |
| Number of generations $n_G$ | Iterations | 50 |
| Selection | Tournament between parents | 3 |
| Crossover type and rate $\chi$ | Single point at the middle | 0.9 |
| Mutation type and rate $\mu$ | Random bit flip | 0.1 |
| Fitness function | Determines members that survives | $F_1$ |

**Algorithm 1** GA($n_G,\chi,\mu$)

```
1: Initialization of P_k = (S, U)        ▷ Population of n randomly-gen-
erated individuals
2: Evaluate P_k        ▷ Compute F_1 for each i ∈ P_k
3: while k < n_G do
4:    Selection: Select the fittest individual from sets of 3.
5:    Crossover: Select χ×n members of P_k; pair them up; produce off-
spring; insert the offspring into P_{k+1}
6:    Mutate: Select μ×n members of Pk + 1; invert a randomly-selected
bit in each;
7:    Evaluate P_{k+1}        ▷ Compute F_1 for each i ∈ P_k
8: return the fittest individual from P_k
```

Table 5 shows the results of GA applied on different settings. Each row represents a single experiment of a specific population size/number of generations combination. The first column contains the population size. The second column shows the number of generations/iterations of the algorithm. The third contains the amount of synthetic curves with transit used on training stage. The fourth column shows the neural network decision threshold. The fifth column contains the values of $F_1$ on each experiment and finally the sixth column shows the number of $F_1$ calculations i.e the amount of models trained in order to get that $F_1$ score value. The best value of $F_1$ (0.9801) is obtained for three different configurations: 1) 10 chromosomes and 50 generations, 2) 20 chromosomes and 20 generations and 3) 50 chromosomes and 10 generations. The three of them have the same value of $S$ (1403), this value corresponds to $\lambda = 74.3\%$ of synthetic lightcurves and 27.7% real curves with transit on training stage. The value of $T$ varies between 0.21 and 0.23 so there is no big difference between the three of them. The configuration 1) was chosen because it is the one that trains the fewest models to obtain the same result and it also has the highest threshold $T$.

## Sensitivity analysis

A fine tunning is performed by analyzing the dependence of the $F_1$ score value on the values of $\lambda$ and $T$. The $\lambda$ value is ranged between 0 and 80%, increasing $S$ from 0 to 1932 in steps of 23. This provides more resolution between $60 \leq \lambda \leq 80\%$ since the best ratio according to the

**Table 5. GA results for different parameter settings.**

| Population | #Generations | S | T | F₁ | #F₁ calc. |
|---|---|---|---|---|---|
| **10** | 5 | 1633 | 0.09 | 0.9607 | 80 |
| | 10 | 1173 | 0.07 | 0.9371 | 160 |
| | 15 | 1334 | 0.41 | 0.9591 | 240 |
| | 20 | 759 | 0.28 | 0.9607 | 320 |
| | **50** | **1403** | **0.23** | **0.9801** | **800** |
| **20** | 5 | 1794 | 0.10 | 0.9560 | 180 |
| | 10 | 1403 | 0.28 | 0.9751 | 380 |
| | 15 | 1403 | 0.19 | 0.9753 | 540 |
| | **20** | **1403** | **0.22** | **0.9801** | **820** |
| | 50 | 1794 | 0.1 | 0.9560 | 1800 |
| **50** | 5 | 1334 | 0.23 | 0.9651 | 440 |
| | **10** | **1403** | **0.21** | **0.9801** | **880** |
| | 15 | 1403 | 0.19 | 0.9753 | 1320 |
| | 20 | 1403 | 0.18 | 0.9705 | 1760 |
| | 50 | 1403 | 0.2 | 0.9753 | 4400 |

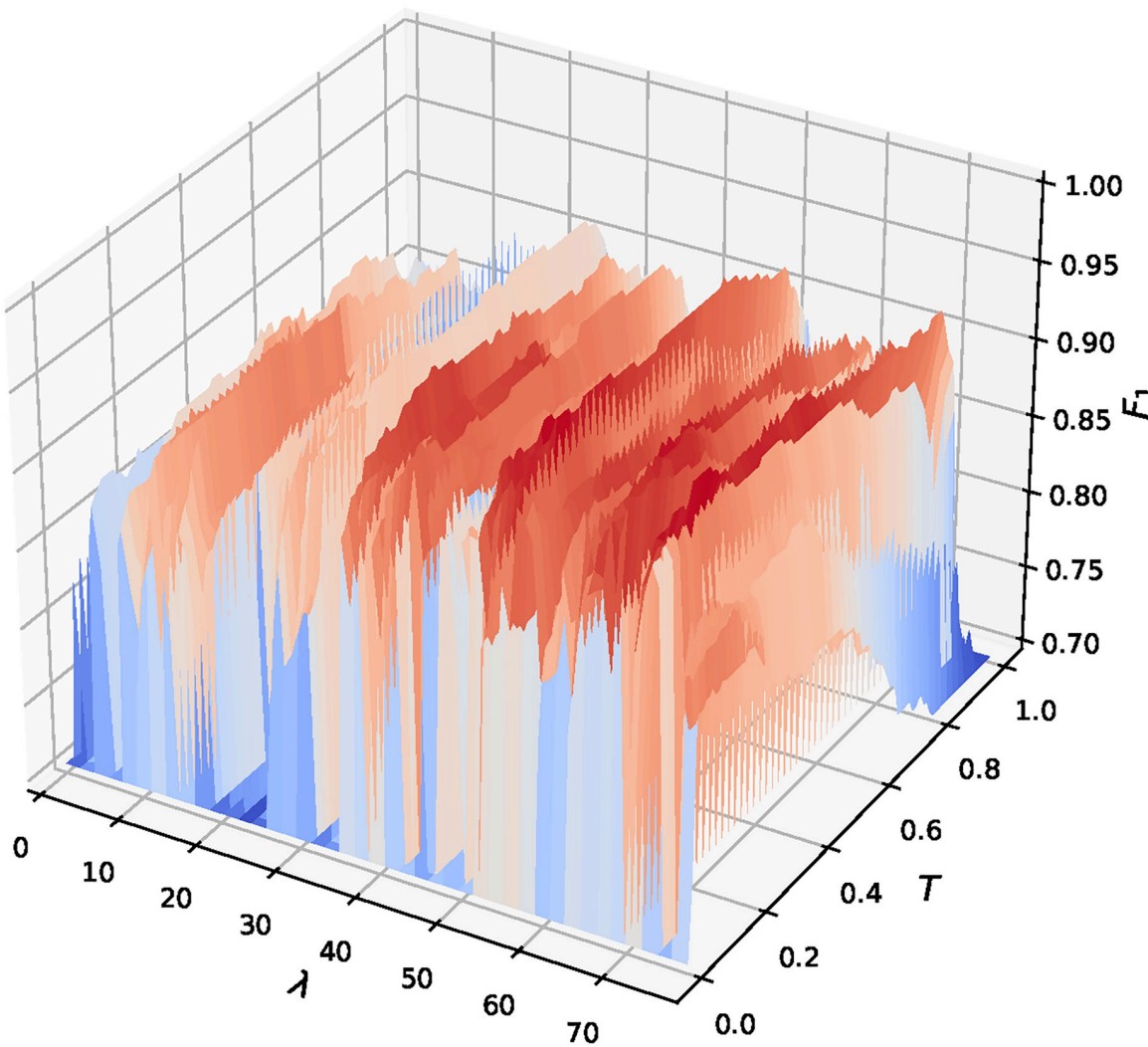

**Fig 3. 3D plot of $F_1$ against ratio λ and threshold $T$.**

previous section is between this range. The threshold value $T$ is ranged from 0 to 1 using two decimals. Each model is trained using $R$ = 483 real lightcurves with transit, $S$ synthetic lightcurves and $S + R$ real lightcurves without transit to mantain balance. The value of $F_1$ is calculated for each model with 200 real lightcurves, half of them with transit and the other half without transit. Fig 3 presents a 3D plot of the $F_1$ score for each pair of λ and $T$ values. It can be observed that for λ greater than 50% the value of $F_1$ is higher, which proves the hypothesis that increasing the number of synthetic lightcurves improves detection performance. It can also be observed that for λ between 72% and 77% and for $T$ between 0.1 and 0.4 the highest $F_1$ values are obtained, with the maximum visible value in λ ≈ 74% and $T$ ≈ 0.2, which is consistent with the optimum value found in the previous section.

To get a broader view of the effect of the decision threshold $T$, models were trained in the same way as the above by ranging the value of λ between 0 and 80% in steps of 5%. For each ratio, the threshold $T$ was varied and TPR and FPR were calculated to construct the Receiver Operating Characteristic Curve (ROC), see Fig 4.

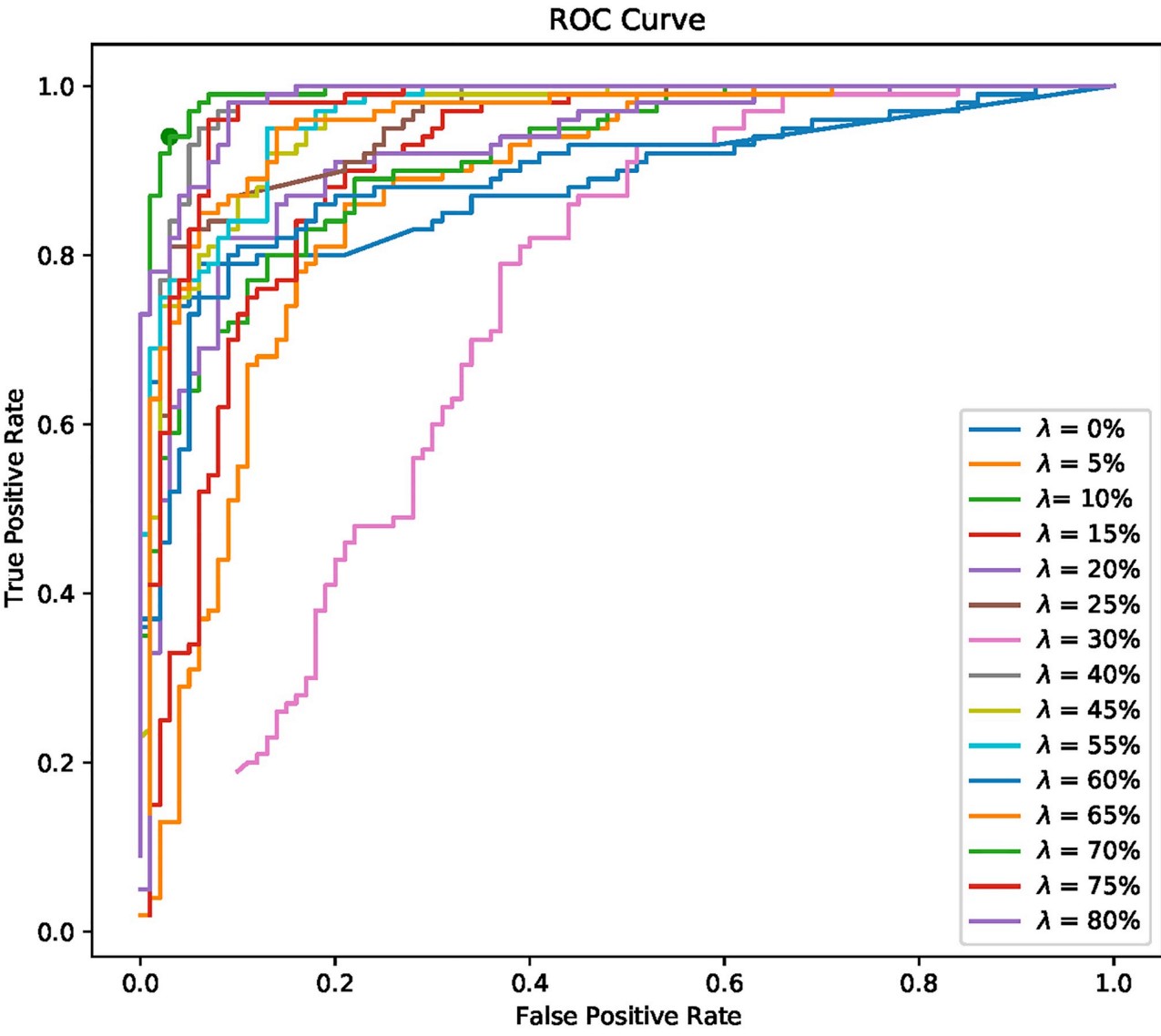

**Fig 4. ROC curve of every ratio.**

It shows that increasing the threshold $T$ increases the rate of true positives, but tends to misclassify negative instances, so the threshold value for which the ROC curve is closest to the ideal case (FPR = 0, TPR = 1) must be found, this value will be denoted as the *best* threshold. It can be seen that the curve of ratio $\lambda = 70\%$ has the point (0.050, 0.970) that is the closest to the ideal one, which is consistent with the value presented in the coarse adjustment with GA.

Table 6 shows the results obtained for each ratio. First column presents the ratio $\lambda$ in percentage. The second column shows the respective number of synthetic light curves $S$. Third column presents the value of the threshold $T$ closer to the ideal ROC curve point. Columns 4 to 9 show the evaluation metrics described on section Evaluation Metrics.

It can be observed that from first limit scenario where $\lambda = 0$ (see Table 3), increasing only the threshold $T$ from 0.5 to 0.608, the precision is improved from 0.747 to 0.923 and thus the $F_1$ value from 0.743 to 0.808.

**Table 6. Sensivity analysis results.**

| λ(%) | S | T | Accuracy | Precision | TPR | FPR | F$_1$ | FNR |
|------|------|-------|----------|-----------|-------|-------|-------|-------|
| 0 | 0 | 0.608 | 0.860 | 0.923 | 0.720 | 0.060 | 0.808 | 0.280 |
| 5 | 25 | 0.639 | 0.825 | 0.825 | 0.850 | 0.210 | 0.837 | 0.150 |
| 10 | 60 | 0.953 | 0.835 | 0.818 | 0.900 | 0.130 | 0.857 | 0.100 |
| 15 | 85 | 0.394 | 0.840 | 0.876 | 0.920 | 0.210 | 0.897 | 0.080 |
| 20 | 121 | 0.631 | 0.870 | 0.922 | 0.830 | 0.080 | 0.873 | 0.170 |
| 25 | 161 | 0.999 | 0.885 | 0.824 | 0.940 | **0.030** | 0.878 | 0.060 |
| 30 | 207 | 0.608 | 0.710 | 0.810 | 0.940 | 0.370 | 0.870 | 0.060 |
| 35 | 260 | 0.456 | 0.850 | 0.873 | **0.970** | 0.070 | 0.919 | **0.030** |
| 40 | 322 | 0.963 | 0.945 | 0.873 | **0.970** | 0.080 | 0.919 | **0.030** |
| 45 | 395 | 0.685 | 0.895 | 0.882 | 0.900 | 0.130 | 0.891 | 0.100 |
| 50 | 483 | 0.578 | 0.840 | 0.897 | 0.880 | 0.040 | 0.888 | 0.120 |
| 55 | 590 | 0.727 | 0.910 | 0.782 | 0.720 | 0.130 | 0.750 | 0.280 |
| 60 | 730 | 0.765 | 0.855 | 0.932 | **0.970** | 0.090 | 0.950 | **0.030** |
| 65 | 897 | 0.444 | 0.905 | 0.938 | 0.910 | 0.140 | 0.923 | 0.090 |
| 70 | 1127 | 0.234 | **0.960** | **0.950** | **0.970** | 0.050 | **0.960** | **0.030** |
| 75 | 1450 | 0.561 | 0.940 | 0.940 | 0.950 | 0.060 | 0.945 | 0.050 |
| 80 | 1932 | 0.141 | 0.945 | 0.872 | 0.960 | 0.090 | 0.914 | 0.040 |

The TPR value, i.e. those light curves with transit that are correctly detected, starts to increase from λ = 25%. This means that adding synthetic curves add knowledge to the model and helps it to more easily identify real curves with transit. On the other hand, the precision starts to increase from λ = 60%. This implies that adding synthetic curves can improve the detection precision, however this generates a bias in the model since it is difficult for it to detect the real light curves with transit, so it is necessary to decrease the decision threshold $T$ that separates the two classes.

## Comparison with related work

The best model was obtained with λ = 74.3% and $T$ = 0.23 (see section Heuristic search of optimal parameters). To evaluate the model with $R$ = 200 real lightcurves (100 with transit, 100 without transit) were used, achieving a precision of 0.9705, a TPR of 0.99, a $F_1$ of 0.9801, a FPR of 0.03 and an accuracy of 0.98. Given the wide range of databases, the comparison between the presented approach and the related work will be centered on two works from Table 1: The one presented in [19] in order to compare two different approaches on the same dataset (Kepler Cumulative Catalog) and the one presented in [24], in order to compare the effect of the same approach on both real and synthetic data.

Table 7 presents the metrics comparison between the proposed approach (ours by short) and the related work described. Column 2 to 5 show the metrics obtained during training ans the rest of them show the metrics obtained during test stage. In the case of [19] which uses real

**Table 7. Comparison of the proposed approach with related work.**

| Ref | Training | | | | Test | | | |
|-----|----------|-----------|-------|-------|----------|-----------|--------|-------|
| | Accuracy | Precision | TPR | F$_1$ | Accuracy | Precision | Recall | F$_1$ |
| [19] | 0.989 | 0.995 | 0.972 | 0.983 | | | | |
| [24] | 1.000 | 1.000 | 1.000 | 1.000 | 0.500 | 0.500 | 0.010 | 0.019 |
| Ours | 0.986 | 0.986 | 0.985 | 0.985 | 0.980 | 0.970 | 0.990 | 0.980 |

data to train the model, the metrics available correspond only to the training stage; this approach does not allow to properly analyse the performance of the model with new data. The expected performance of this model evaluated on unknown real lightcurves should be the one from first scenario presented on Table 3.

On the other hand, the article [24] presented validation metrics but the model is trained and evaluated on synthetic light curves only. In order to perform a fair comparison between our approach and this work and since it contains open source code and a full description of the method available, the model in [24] has been reproduced and evaluated with real light-curves. A precision of 0.5, a TPR of 0.01, a $F_1$ of 0.0196, a FPR of 0.01, and an accuracy of 0.5 was obtained. This performance is very similar than the second scenario presented on Table 3 where the model is unable to detect a real lightcurve with transit since the real transits may not have such a relevant drop in the flux. However the higher value of the precision shows that the proposed approach (i.e., adding synthetic lightcurves to the training stage) brings knowledge to the model. In this case the method demonstrate that combining real and synthetic light-curves on the training stage can improve the detection metrics.

## Conclusion

In this paper, the development of a deep learning system for detecting planetary transits in Kepler Telescope light-curves is presented. The approach is based on related work from the literature and enhanced to validation with real lightcurves. 2D phase folding is used as a feature extraction method that allows real and synthetic light-curves with transit to be described by an image distinguishable from those without transit. The model parameters are adjusted to improve the performance of the classification. The method is evaluated on real light-curves from the Kepler's catalog and demonstrates superior performance against other approaches presented on the state of art.

The main contribution of this work is the enhance of a detection model including the generation of synthetic light-curves with transit from estimated parameters. The best ratio of synthetic data is founded using a coarse tunning with Genetic algorithms and evidenced with a sensibility analysis. The evaluated metrics demonstrate that the combination of real and synthetic light-curves with transit on the training stage add knowledge to the model and improve the performance on real light curves.

Future work will consider extend the study to systems with more than one confirmed planet or planetary candidate dealing with multi-transit detection on the same light-curve. Also the implementation of the method on a different database like the NASA's Transiting Exoplanet Survey Satellite (TESS), mission that has discover already 166 exoplanets and has 4604 planet candidates, or even the data acquired from the James Webb Space Telescope launched on December 2021.

## Author Contributions

**Conceptualization:** Sara Cuéllar, Paulo Granados, Michel Curé, Héctor Vargas, Sebastián Dormido-Canto, Gonzalo Farias.

**Data curation:** Sara Cuéllar.

**Formal analysis:** Sara Cuéllar, Paulo Granados, Ernesto Fabregas, Michel Curé, Sebastián Dormido-Canto, Gonzalo Farias.

**Funding acquisition:** Michel Curé, Héctor Vargas, Sebastián Dormido-Canto, Gonzalo Farias.

**Investigation:** Sara Cuéllar, Paulo Granados, Ernesto Fabregas, Michel Curé, Sebastián Dormido-Canto, Gonzalo Farias.

**Methodology:** Ernesto Fabregas, Michel Curé, Sebastián Dormido-Canto, Gonzalo Farias.

**Project administration:** Ernesto Fabregas, Michel Curé, Sebastián Dormido-Canto, Gonzalo Farias.

**Resources:** Héctor Vargas, Gonzalo Farias.

**Software:** Sara Cuéllar.

**Supervision:** Ernesto Fabregas, Michel Curé, Sebastián Dormido-Canto, Gonzalo Farias.

**Validation:** Sara Cuéllar, Héctor Vargas, Gonzalo Farias.

**Visualization:** Sara Cuéllar, Paulo Granados.

**Writing – original draft:** Sara Cuéllar, Ernesto Fabregas, Héctor Vargas.

**Writing – review & editing:** Ernesto Fabregas, Michel Curé, Héctor Vargas, Sebastián Dormido-Canto, Gonzalo Farias.

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
