## [Decision Letter · Decision Letter 0]

21 Feb 2022

PONE-D-22-01834Deep Learning Exoplanets Detection by Combining Real and Synthetic DataPLOS ONE

Dear Dr. Fabregas,

Thank you for submitting your manuscript to PLOS ONE. After careful consideration, we feel that it has merit but does not fully meet PLOS ONE’s publication criteria as it currently stands. Therefore, we invite you to submit a revised version of the manuscript that addresses the points raised during the review process.

We look forward to receiving your revised manuscript.

Kind regards,

Sathishkumar V E

Academic Editor

PLOS ONE

Journal Requirements:

"This research was supported in part by the Chilean Ministry of Education under Projects FONDECYT 1191188 and 1190486"

"This research was supported in part by the Chilean Ministry of Education under Projects FONDECYT 1191188 and 1190486."

Reviewers' comments:

Reviewer's Responses to Questions

**Comments to the Author**

1. Is the manuscript technically sound, and do the data support the conclusions?

Reviewer #1: Partly

Reviewer #2: Yes

2. Has the statistical analysis been performed appropriately and rigorously? 

Reviewer #1: No

Reviewer #2: No

3. Have the authors made all data underlying the findings in their manuscript fully available?

Reviewer #1: Yes

Reviewer #2: Yes

4. Is the manuscript presented in an intelligible fashion and written in standard English?

Reviewer #1: Yes

Reviewer #2: Yes

5. Review Comments to the Author

Reviewer #1: 1. The authors have to list the research questions addressed by the proposed work and also the need for the proposed research

2. Xception architecture is used. But the authors did not explain the rationale behind choosing this model among many CNN models.

3. the need for and applications of the proposed work to be discussed

Reviewer #2: 1. "The best ratio

of synthetic data is determined by the perform of an optimization technique and a

sensitivity analysis." Check grammar

2. Highlight the the proposed methods significance with quantitative results

3. "Section presents some exoplanet detection 67

approaches that can be found in the literature and describes briefly the approach which 68

is the start point of this work. Section details the proposed method. Section shows the 69

experimental results and a comparison with previous results. Finally, Section 70

summarizes the main conclusions and future work." Section numbers are missing

3. Justification for synthetic data generation need to be elaborately given

4. Table 4 "Encode S" description is missing

5. Pseudocode of GA related to the given problem has to be included with explanation

6. Only GA is used for comparison. How does other meta-heuristic algorithm's like PSO, ACO, BA etc., perform?

7. Description about the CNN model used is required

8. Citation to the figures 1 and 2 is missing.

6. PLOS authors have the option to publish the peer review history of their article (what does this mean?). If published, this will include your full peer review and any attached files.

Reviewer #1: No

Reviewer #2: No

---

## [Author Response · Author response to Decision Letter 0]

31 Mar 2022

Response to Reviewer 1 Comments

Point 1: The authors have to list the research questions addressed by the proposed work and also the need for the proposed research.

Response 1: Thank you for this comment. The main research questions addressed by this work are: How can artificial intelligence-based algorithms contribute to the exoplanet detection field, also if it is possible to add technical knowledge through synthetic data to improve the performance of the exoplanet detector.

The need of this research relies on three main aspects: First, the exoplanet detection as an opportunity to explore areas to look for other habitable worlds. Second, given many databases, the use of artificial intelligence to analyse massive data. Finally, the gaps that it aims to fill in the literature based on the previous works. Some examples of those gaps are the use of unconfirmed planets to train the models and the lack of evaluation of the models on unknown real data. To clarify this point, we modified the introduction.

Point 2: Xception architecture is used. But the authors did not explain the rationale behind choosing this model among many CNN models.

Response 2: Thank you for this comment. To clarify this point, we modified the detection model subsection.

Point 3: The need for and applications of the proposed work to be discussed.

Response 3: Thank you for this comment. The applications and future work extend the study to systems with more than one confirmed planet or planetary candidate dealing with multi-transit detection on the same light curve. Also, the implementation of the method on a different database like NASA’s Transiting Exoplanet Survey Satellite (TESS), the mission has discovered already 450 166 exoplanets and has 4604 planet candidates or even the data acquired from the James Webb Space Telescope launched in December 2021. To clarify this point, we modified the introduction and conclusion.

Response to Reviewer 2 Comments

Point 1: "The best ratio of synthetic data is determined by the performance of an optimization technique and a sensitivity analysis." Check grammar.

Response 1: Thank you for this comment. To clarify this point, we modified the abstract. We also performed a complete check of the paper and asked for a grammar, format, and style revision from experts.

Point 2: Highlight the significance of the proposed method with quantitative results.

Response 2: Thank you for this comment. The experimental results section performs three quantitative analyses: A heuristic approach to find a small range to narrow down the search for the amount of synthetic data that improves the detection model. Second, a fine adjustment to get the actual values of S and T. Finally, a metric comparison between our approach and the work presented in the literature. In order to clarify this point, we modified the experimental results section.

Point 3: "Section presents some exoplanet detection 67 approaches that can be found in the literature and briefly describe the approach which 68 is the start point of this work. Section details the proposed method. The section shows the 69 experimental results and a comparison with previous results. Finally, Section 70 summarizes the main conclusions and future work." Section numbers are missing.

Response 3: Thank you for this comment. To clarify this point, we modified the paper overview in the introduction.

Point 4: Justification for synthetic data generation need to be elaborately given.

Response 4: Thank you for this comment. The main purpose of synthetic data generation is to take advantage of the technical knowledge acquired from experts of the periodicity and modelling of the planet's transit and translate it into synthetic data that can be added to the model. To clarify this point, we modified the synthetic data generation subsection.

Point 5: Table 4 "Encode S" description is missing.

Response 5: Thank you for this comment. To clarify this point, we modified the Table 4 content.

Point 6: The pseudocode of GA related to the given problem must be included with an explanation.

Response 6: Thank you for this comment. We add the pseudocode of GA to the experimental results section.

Point 7: Only GA is used for comparison. How do other meta-heuristic algorithms like PSO, ACO, BA etc., perform?

Response 7: The main purpose of the heuristic search on this approach was to find a small range to narrow down the search for the values of S and T to obtain the highest possible F1 value. Those values are used to perform a fine adjustment described in the fourth section. We select GA based on its great performance and easy implementation. Given the nature of the optimization problem, we expect that other meta-heuristic algorithms find the same range of S and T. To clarify this issue, we modified the GA description.

Point 8: Description of the CNN model used is required.

Response 8: Thank you for this comment. To clarify this point, we modified the Detection model subsection.

Point 9: Citation to figures 1 and 2 is missing.

Response 9: Thank you for this comment. We already checked and both figures are cited in the text.

---

## [Decision Letter · Decision Letter 1]

25 Apr 2022

Deep Learning Exoplanets Detection by Combining Real and Synthetic Data

PONE-D-22-01834R1

Dear Dr. Fabregas,

We’re pleased to inform you that your manuscript has been judged scientifically suitable for publication and will be formally accepted for publication once it meets all outstanding technical requirements.

Kind regards,

Sathishkumar V E

Academic Editor

PLOS ONE

Additional Editor Comments (optional):

Reviewers' comments:

Reviewer's Responses to Questions

**Comments to the Author**

1. If the authors have adequately addressed your comments raised in a previous round of review and you feel that this manuscript is now acceptable for publication, you may indicate that here to bypass the “Comments to the Author” section, enter your conflict of interest statement in the “Confidential to Editor” section, and submit your "Accept" recommendation.

Reviewer #1: All comments have been addressed

Reviewer #3: All comments have been addressed

2. Is the manuscript technically sound, and do the data support the conclusions?

Reviewer #1: Partly

Reviewer #3: (No Response)

3. Has the statistical analysis been performed appropriately and rigorously? 

Reviewer #1: Yes

Reviewer #3: Yes

4. Have the authors made all data underlying the findings in their manuscript fully available?

Reviewer #1: Yes

Reviewer #3: Yes

5. Is the manuscript presented in an intelligible fashion and written in standard English?

Reviewer #1: Yes

Reviewer #3: Yes

6. Review Comments to the Author

Reviewer #1: The authors may add a paragraph to explain how they have addressed the research questions through their proposed work.

Reviewer #3: The authors addressed all the recommended comments and the current version of the paper is well improved. This version of the paper is recommended for publication

7. PLOS authors have the option to publish the peer review history of their article (what does this mean?). If published, this will include your full peer review and any attached files.

Reviewer #1: No

Reviewer #3: No

---

## [Editor Report · Acceptance letter]

5 May 2022

PONE-D-22-01834R1 

Deep Learning Exoplanets Detection by Combining Real and Synthetic Data 

Dear Dr. Fabregas:

I'm pleased to inform you that your manuscript has been deemed suitable for publication in PLOS ONE. Congratulations! Your manuscript is now with our production department. 

Kind regards, 

on behalf of

Dr. Sathishkumar V E 

Academic Editor

PLOS ONE